# Unprecedented switching endurance affords for high-resolution surface temperature mapping using a spin-crossover film

Karl Ridier [1], Alin-Ciprian Bas[1], Yuteng Zhang[1], Lucie Routaboul[1], Lionel Salmon[1], Gábor Molnár[1✉], Christian Bergaud[2] & Azzedine Bousseksou[1✉]

Temperature measurement at the nanoscale is of paramount importance in the fields of nanoscience and nanotechnology, and calls for the development of versatile, high-resolution thermometry techniques. Here, the working principle and quantitative performance of a cost-effective nanothermometer are experimentally demonstrated, using a molecular spin-crossover thin film as a surface temperature sensor, probed optically. We evidence highly reliable thermometric performance (diffraction-limited sub-μm spatial, μs temporal and 1 °C thermal resolution), which stems to a large extent from the unprecedented quality of the vacuum-deposited thin films of the molecular complex [Fe(HB(1,2,4-triazol-1-yl)$_3$)$_2$] used in this work, in terms of fabrication and switching endurance (>$10^7$ thermal cycles in ambient air). As such, our results not only afford for a fully-fledged nanothermometry method, but set also a forthcoming stage in spin-crossover research, which has awaited, since the visionary ideas of Olivier Kahn in the 90's, a real-world, technological application.

---

[1] Laboratoire de Chimie de Coordination, CNRS UPR 8241, 205 route de Narbonne, F–31077 Toulouse, France. [2] Laboratoire d'Analyse et d'Architecture des Systèmes, CNRS UPR 8001, 7 avenue du Colonel Roche, F–31400 Toulouse, France. ✉email: gabor.molnar@lcc-toulouse.fr; azzedine.bousseksou@lcc-toulouse.fr

The recent achievements in nanoscience and nanotechnology brought about the growing need for nanothermometry techniques capable of measuring temperature on a reduced size scale and often during limited time. In particular, the highly increased density of electronic components per unit area, together with miniaturization and high-frequency operation, can lead to localized (over)heating problems (sic "hot spots"), making thermal management a key factor for determining the performance and the reliability of integrated circuits[1]. As such, the development of high spatial and temporal resolution thermometry techniques appears as critical in order to understand the local thermal processes at work as well as to improve the design of nanoscale devices[2–4].

To reach sub-wavelength spatial resolution, the most promising techniques are those involving scanning transmission electron microscopy (STEM)[5] or scanning probe microscopy (SPM), such as scanning thermal microscopy (SThM)[6–10]. Although these techniques can afford for temperature measurements down to sub-10 nm length scale[5,9], they are often invasive, require sophisticated equipment and are difficult to implement or transpose for everyday practical applications. In addition, their point-by-point data acquisition process inherently results in slow imaging capabilities.

Contactless far-field optical thermometry techniques, including infrared thermography (IRT)[11], thermoreflectance microscopy[12–14], optical interferometry[15], or Raman spectroscopy[16,17], represent to some extent a compromise between overall thermometric performance and general applicability. These techniques usually provide fast (typically ns–µs) thermal-imaging capabilities with diffraction-limited spatial resolution, and represent the industry standard in terms of surface thermal imaging[18,19]. Nevertheless, a recurrent issue of these different techniques is the requirement for extensive, surface-dependent calibration procedures (emissivity, thermoreflectance coefficient, thermal expansion coefficient, etc.) to arrive at a quantitative determination of the temperature.

Another interesting approach involves the surface deposition of a temperature-sensitive material, which can be probed remotely, most often through optical methods. A well-known example is luminescent thermometry[2,20], which is based on the temperature-dependent emission intensity and/or lifetime of the emitting state of luminophore species[21–23]. Another example is the commercially available liquid crystal thermography (LCT) technique[24], for which the local temperature is inferred from the selective light reflection of a surface coated with thermochromic liquid crystal materials.

However, these techniques can see their performance deteriorate rapidly, mainly due to (photo)stability issues. Although these surface-coating methods can be classified as semi-invasive because they may involve a potential disturbance in the temperature field, the integration of temperature-sensitive phase-change materials (PCMs) appears as a promising approach, offering a multitude of possibilities in terms of thermal sensitivity, thermometric properties, and readout[25–28].

In this context, molecular spin-crossover (SCO) materials constitute a promising class of PCM for surface thermometry and thermal-imaging applications. These compounds, which exhibit a reversible solid–solid transition between the so-called low-spin (LS, low-temperature) and high-spin (HS, high-temperature) electronic configurations[29–31], are known to display a drastic change of their optical properties (absorption, refractive index) under the effect of a temperature variation. Interestingly, different types of application can be envisaged according to the characteristics of the thermal spin transition. A SCO thin film exhibiting a thermal hysteresis loop can be used for thermal memory applications[32], but, a contrario, the absence of hysteresis allows for real-time thermometry. Although first proofs of concept have been reported[32,33], the realization of a real-world SCO-based application was primarily hampered by the weak reliability and the recurrent problem of mechanical fatigue of SCO materials upon extended switching, the SCO cycling being typically restricted to only few/tens/hundreds thermal cycles before failure. This issue has constituted so far, the long-standing blocking point for SCO molecules.

In this study, we show that the molecular SCO complex [Fe(HB(tz)₃)₂] (tz = 1,2,4-triazol-1-yl) **1**, recently synthesized in the form of high-quality, vacuum-deposited thin films[34], overcomes this major bottleneck. This compound exhibits an exceptionally high spin-state switching endurance and long-term stability upon repeated thermal cycling. Taking advantage of this unprecedented feature, together with its remarkable spin-transition properties (abrupt SCO above room temperature), we experimentally demonstrate that thin films of **1** can be used as a high-performance temperature sensor, probed by optical reflectivity, for visualizing and measuring the surface temperature. The capabilities of this SCO-based thermal microscopy technique are quantified on a series of Joule-heated metallic nanowires. Notably, the possibility of mapping the surface temperature distribution with a sub-µm spatial, µs temporal, and 1 °C thermal resolution is demonstrated.

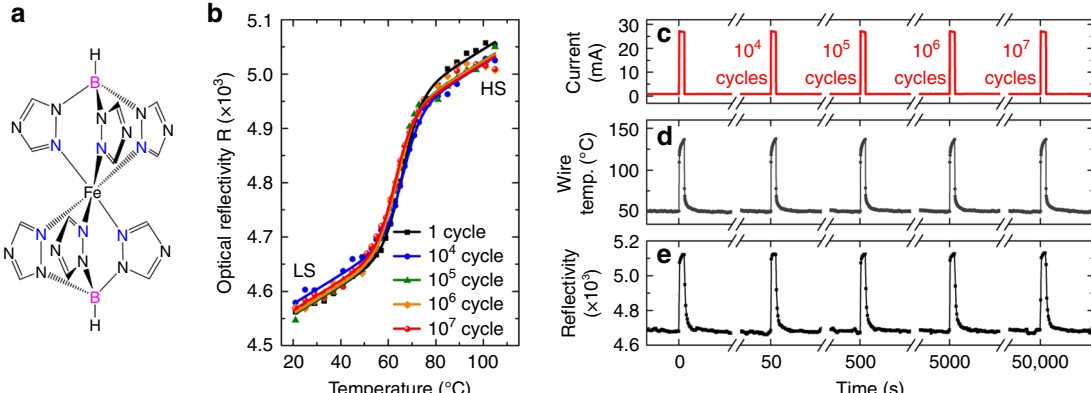

**Fig. 1 Phase-change properties and switching endurance of the thin films of 1. a** Molecular structure of complex **1**. **b** Thermal evolution of the optical reflectivity ($\lambda = 452$ nm) of a 200-nm-thick film of **1** on top of a gold microwire (2 mm × 2 µm × 300 nm) after 1, $10^4$, $10^5$, $10^6$, and $10^7$ thermal cycles in air induced by Joule heating. These data show the large variation of the reflectivity due to the molecular spin-state switching and demonstrate the exceptionally high resilience of the spin transition over $10^7$ thermal cycles. Time evolution of **c** the applied electrical current, **d** the wire temperature, and **e** the optical reflectivity on the wire recorded simultaneously at selected times of the thermal-cycling experiment using the Joule-heated microwire.

## Results

**Spin-transition properties and switching endurance of 1.** Compound **1** is a molecular SCO complex[35] (Fig. 1a) which can be deposited in the form of high-quality, large-area, nanometer-thick films by vacuum thermal evaporation[34]. The resulting nanocrystalline films are known to be oriented (with the *c*-axis normal to the substrate plane), homogenous, and they display a fully complete, isostructural spin transition centered at $T_{SCO} =$ 63–65 °C (with a slight dependence upon the film thickness)[34,36]. Importantly, the thermal spin transition in the thin films of **1** is known to be relatively abrupt, i.e. 80% of the transition is spanned over a temperature range of ca. 12 °C, with a narrow (<0.5 °C-wide) thermal hysteresis loop. The SCO phenomenon is accompanied by a drastic change of the optical properties of the films, in particular of the UV absorbance (Supplementary Fig. 1) as well as the optical reflectivity (*R*) on various surfaces (Fig. 1b and Supplementary Fig. 2). The optical reflectivity measured at $\lambda = 452$ nm on a 200-nm-thick film of **1**, deposited on top of a gold surface, typically increases by ca. 7% due to the switching from the LS to the HS state, while it goes down on glass (−4%) and silicon (−1.5%).

An important and quite unexpected discovery with compound **1** is the existence of an exceptionally high stability and robustness of the spin-transition properties, as well as a strong resilience of the material upon extended thermal cycling. First, from UV absorption measurements, we demonstrated that the transition temperature, the shape, and the completeness of the thermal spin transition in the thin films of **1** remain virtually unaltered over a period of more than 1 year of storage in ambient air (Supplementary Fig. 3). These optical measurements have also revealed that the SCO properties are fully preserved after annealing the films at temperatures as high as 230 °C (Supplementary Fig. 3). To investigate the switching endurance of thin films of **1**, we first performed successive thermal cycles using a heating/cooling Peltier stage, while the spin-transition properties were probed through optical absorbance measurements. Using this time-consuming "classical" approach, the switching properties of the films are found to be preserved after more than 10,000 thermal cycles in air (see Supplementary Fig. 1). Critically, the transition temperature shifts <0.5 °C.

To go further, we used Joule-heated gold microwires as fast micro-heaters, to expose a thin film of **1** to a large number of thermal cycles ($10^7$) within a reasonable time (<14 h). These microwires, fabricated on top of a glass substrate by means of conventional photolithography (PL) methods (see "Methods" section and Supplementary Fig. 4), were coated with a 200-nm-thick film of **1** and mounted on a variable-temperature microscope stage to control the temperature of the entire substrate. This latter, which differs from the actual temperature of the heating wire $T_{wire}$, will be hereafter denoted "base temperature" $T_b$. Due to their small thermal mass, these microwire heaters are known to be particularly interesting for generating fast (<µs), localized heating (*T*-jumps)[32,37]. Besides, the time-resolved measurement of their electrical resistance, using a custom-built differential resistance measurement setup[37], offers the possibility of monitoring in situ the average temperature rise of the wire with a sub-µs time resolution (Supplementary Fig. 4). For our thermal-cycling experiment, the temperature of the wire was modulated between 50 and 88 °C by applying 200-µs-long current pulses at a frequency of 200 Hz. Note that the heating duration (200 µs) was chosen long enough compared to the thermally activated molecular spin-state switching time (a few tens of nanoseconds above room temperature[38]), ensuring that a full LS–HS–LS thermal switching cycle was completed for each current pulse. At selected times, the excitation frequency was lowered to 0.2 Hz in order to

record simultaneously the temporal evolution of the optical reflectivity ($\lambda = 452$ nm) on the heated microwire. As depicted in Fig. 1c–e, significant jumps of the reflectivity signal are observed, confirming the occurrence of the spin-state switching event following the oscillating excitation current. Crucially, the amplitude of the reflectivity oscillations remains comparable even after $10^7$ cycles, demonstrating the exceptional robustness of the spin transition in the thin films of **1**. At different time intervals of the thermal-cycling experiment, the reflectivity signal on the SCO-coated wire was also acquired as a function of the base temperature (Fig. 1b). Remarkably, the spin-transition curves (switching temperature, shape, and amplitude) remain virtually unchanged after more than 10 million thermal-switching events.

This is the first time that such an outstanding resilience is demonstrated for a SCO compound. It is important to mention that the spin-transition properties of **1** remain unaltered despite a relatively large volume change (ca. 4.5%) upon the SCO[35]. Indeed, as with many PCMs, the pivotal issue is the large transformation strain, which leads to the build-up of mechanical stress and, ultimately, to material failure. This inherent fatigability currently constitutes a common hurdle for the implementation of various PCMs and molecular switches into demanding technological applications[39–41]. The exceptionally high switching endurance of thin films of **1** must probably arise from a combination of structural factors. First, the molecular nature of compound **1** (with only weak intermolecular bonds) and its processing in the form of high-purity (vacuum-deposited) thin films are undoubtedly important features that minimize the transformational stress during the spin-state switching. Most importantly, as demonstrated for shape memory alloys[40,42], and also suggested for $VO_2$[43], the overall crystallographic compatibility of the two phases appears as a key parameter to achieve high switching endurance. In **1**, this compatibility includes the isostructural character of the transition (same orthorhombic space group *Pbca* in the two phases) as well as a pronounced anisotropy of the transformation strain, the structural deformation mainly occurring along the *c*-axis ($\Delta c/c = +5.6\%$)[35]. This anisotropic deformation allows the crystal lattice to better accommodate the volume change, through the possible formation of mismatch-free LS/HS phase boundaries during the spin-state switching[44]. On the other hand, as the thin films of **1** are oriented (with the *c*-axis normal to the substrate), the transformation strain is almost zero in the substrate plane[45] and the spin transition thus gives rise to rather small mechanical stress in the film. While more comprehensive studies still need to be conducted on these structure–property relationships, the negligible fatigability of **1**, together with its remarkable spin-transition properties, make this compound an ideal candidate for high-cycle applications, such as thermal imaging.

**Working principles of the SCO-based surface thermometer.** To demonstrate the capabilities of our SCO-based thermometer, we used electron-beam lithography (EBL) to fabricate a variety of Joule-heated gold nanowires (80-µm-long, 50-nm-thick) on glass and oxidized silicon chips (see Fig. 2 and Supplementary Fig. 5). As already demonstrated[32,37], their fast temporal response with their reliable all-electrical operation make these wires ideal test benches for the development of nanothermometry applications. The chips were fully covered with a 200-nm-thick film of **1**, grown by vacuum thermal evaporation. It is important to mention that both the deposition conditions as well as the physical properties of the SCO thin film are not invasive and do not modify the overall electrical and thermal performance of the devices (vide infra).

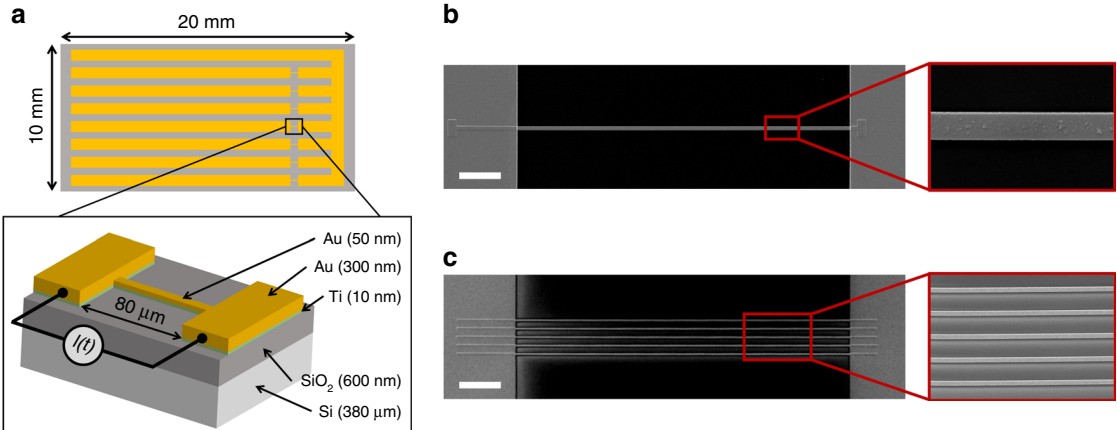

**Fig. 2 Description of the Joule-heated gold nanowires used for nanothermometry. a** Sketch of the chips used (each comprising seven 80-μm-long, 50-nm-thick gold wires), and schematic representation of a nanowire on top of an oxidized silicon substrate. Scanning electron microscopy (SEM) images of **b** a 1-μm-wide wire and **c** five parallel 500-nm-wide wires spaced 1.5 μm apart. Scale bars, 10 μm.

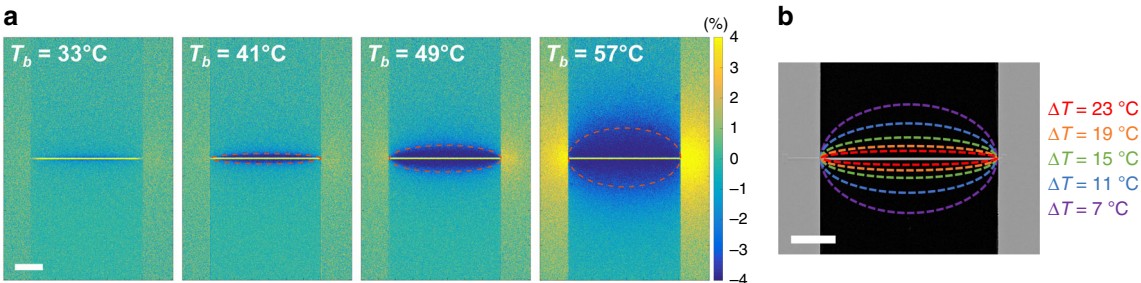

**Fig. 3 Contour map of the temperature distribution induced by a Joule-heated nanowire.** The wire ($80\,\mu m \times 1\,\mu m \times 50\,nm$) on top of a glass substrate is coated with a 200-nm-thick film of **1** and Joule-heated by the application of an electrical current of 4 mA. **a** Optical reflectivity ($\lambda = 452\,nm$) images $\Delta R/R = (R_{ON}-R_{OFF})/R_{OFF}$ of the nanowire recorded at selected base temperatures $T_b$. The dotted lines indicate the position of the corresponding isotherm ($T_{SCO} = 64\,°C$). **b** Contour map of the surface temperature acquired from gathering the isotherms for various base temperatures ($\Delta T = T_{SCO} - T_b$). Scale bars, 20 μm.

While a thin film exhibiting a gradual SCO enables continuous temperature monitoring[33] (provided that a careful thermal calibration of the measured physical properties is performed), here we suggest another approach. Taking advantage of the abrupt character of the thermal spin transition, we demonstrate that thin films of **1** can be used as a binary sensor of the surface temperature, because a sizeable jump of the optical reflectivity occurs at the transition temperature $T_{SCO} = 64\,°C$, bringing out a well-defined isotherm. This property can be advantageously used to record fast contour maps of the surface temperature distribution. As shown in Fig. 3a, reflectivity images of a Joule-heated nanowire on glass were recorded by optical microscopy in both ON ($I \neq 0$) and OFF ($I = 0$) states of the heating device at various base temperatures $T_b$ of the glass substrate, such that $\Delta R/R = (R_{ON} - R_{OFF})/R_{OFF}$ images could be obtained for each $T_b$. (Supplementary Fig. 6 displays a more complete series of images.) On these images, two distinct areas are clearly discernible. One region, close to the nanowire, where the optical reflectivity has changed following the application of the electrical current, signaling the thermal switching into the HS state. Beyond this region, the optical reflectivity remains basically unchanged. As displayed in Fig. 3a, these two areas are separated by an isothermal line, with an elliptical shape on glass, on which the temperature rise is simply given by the difference $\Delta T = T_{SCO} - T_b$. Heating or cooling the chip at different base temperatures makes it possible to acquire a thermal contour map with as many isotherms as desired (Fig. 3b). This simple protocol turns out to be extremely powerful for determining temperature gradients in a

quick manner, or for locating, at first glance, the existence of hot spots on the surface (vide infra). A great advantage is that fast thermal mapping can be achieved without any thermal calibration of surface properties.

A more accurate method to determine the local heating (for example for each pixel of the CCD camera) is based on the comparison of the thermal transition curves, inferred from the measurement of the optical reflectivity signal over a large range of $T_b$, in both ON and OFF states of the heating device. As shown in Fig. 4a, we applied this procedure to map the temperature distribution of the same Joule-heated nanowire on glass. As an example, Fig. 4c displays typical ON-state and OFF-state transition curves obtained for selected pixels of the reflectivity images at different distances from the wire. For each pixel, the value of the local heating was simply deduced from the thermal shift ($\Delta T = T_{SCO}^{OFF} - T_{SCO}^{ON}$), by way of an automated curve-fitting procedure using sigmoidal functions. As shown in Fig. 4a, this procedure allows to map the temperature field with a micrometer spatial resolution, incidentally revealing a slightly inhomogeneous heating of the nanowire. As deduced from the thermal map, the application of a current of 4 mA induces an average heating of $\Delta T_{wire} = 40.4 \pm 0.5\,°C$ on the wire, while the temperature rise is $\Delta T = 5.8 \pm 0.3\,°C$ at 50 μm from the center of the wire. (Supplementary Fig. 7 shows experimental and simulated temperature maps with the corresponding cross-sections for different applied electrical currents.)

Following the same procedure, Fig. 4d displays the temperature map ($\Delta T$) of an identical nanowire on top of a silicon substrate,

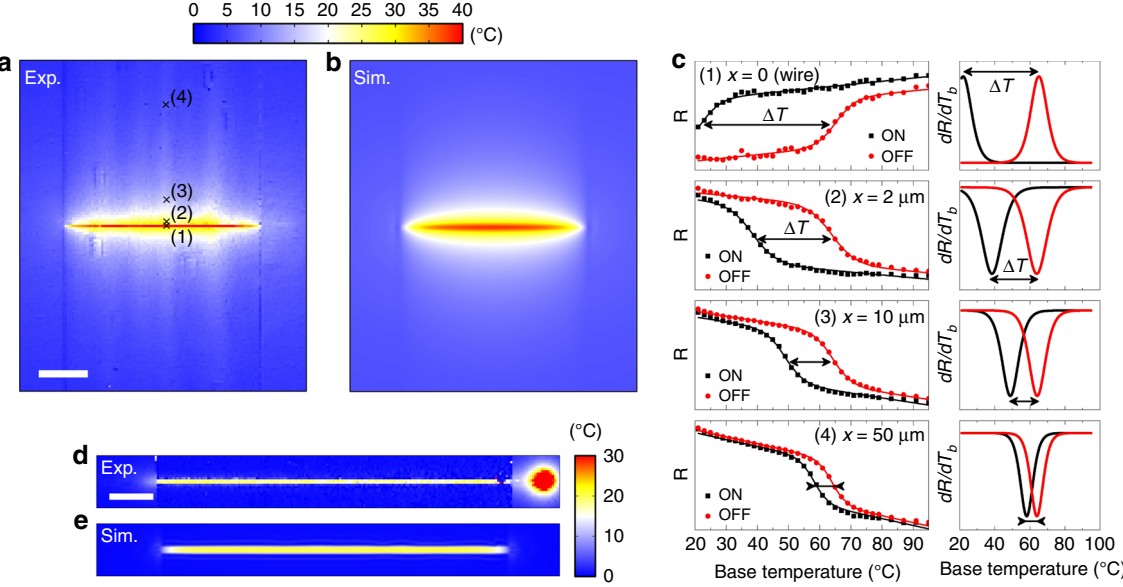

**Fig. 4 Thermal mapping of a Joule-heated gold nanowire on glass and silicon substrates. a** Experimental and **b** simulated temperature maps ($\Delta T$) of the nanowire (80 μm × 1 μm × 50 nm) on glass, heated by the application of an electrical current of 4 mA. The pixel size of the experimental map is 1 × 1 μm$^2$. Scale bar, 20 μm. **c** Typical ON-state ($I = 4$ mA) and OFF-state ($I = 0$) reflectivity curves measured in selected pixels, marked by crosses in **a**, at different distances ($x = 0, 2, 10$, and 50 μm) from the wire. Note that the reflectivity signal increases with $T_b$ on the gold wire, but it goes down on glass. Solid lines are the results of a curve-fitting procedure using a sigmoidal function. The temperature shift between the two peaks in the corresponding derivative curves ($dR/dT_b$ vs. $T_b$) directly gives the average temperature rise in each pixel. **d** Experimental and **e** simulated temperature maps ($\Delta T$) of an identical nanowire on top of a silicon substrate, Joule-heated by the application of an electrical current of 7 mA. The pixel size of the experimental map is 0.5 × 0.5 μm$^2$. Scale bar, 10 μm.

heated by the application of an electrical current of 7 mA. (See Supplementary Fig. 8 for temperature maps and associated cross-sections at different excitation currents.) As expected due to its high thermal conductivity, the silicon behaves as a powerful heat sink. The result is that the heating is predominantly localized on the 1-μm-wide wire, whereas the surrounding Si substrate undergoes almost no temperature rise. The high thermal conductivity of silicon has also the effect of limiting the average temperature rise experienced by the nanowire ($\Delta T_{wire} = 19.8 \pm 0.6$ °C at 7 mA). Another interesting observation is the emergence of a micrometric hot spot at the connection between the wire and the gold electrode (see Fig. 4d), certainly due to a degrading electrical contact. This example nicely illustrates the capabilities of our technique for localizing, in a straightforward manner, micrometric hot spots on electronic circuits.

Our experimental results were validated by finite-element simulations. (N.B. Comparison with other experimental methods is not relevant here as they involve similar, or higher, uncertainty than our approach. Notably, electrical resistance measurements on the nanowires turned out to be ill-reproducible on such small wires.) Simulations were carried out using material properties implemented in the COMSOL program (see "Methods" section). The value of the electrical resistivity of gold, which is the only adjustable parameter, was fixed from the mean electrical resistance of the nanowires measured at room temperature ($R_e \sim 160$ Ω). As shown in Fig. 4b, e (see also Supplementary Figs. 7 and 8), the quantitative agreement between the simulated and experimental data is excellent, both on silicon and glass substrates, for the different excitation currents used.

We took benefit from these finite-element simulations to assess the possible disturbance in the temperature field caused by the presence of the SCO layer. Under (quasi)static conditions—typically reached within the microsecond time scale (vide infra)—we find that the surface temperature differs by <1 °C due to the presence of the SCO layer (Supplementary Fig. 9). This value

remains comparable with the overall measurement uncertainty. Obviously, under non-thermal-equilibrium conditions, the temperature difference between the heating element and the thin-film surface is expected to be larger. In particular, in addition to the "normal" heat capacity of the SCO film, an excess heat capacity should be also considered due to the endothermic character of the LS-to-HS transition ($\Delta H = 40$ J g$^{-1}$ in **1**)[35], which can give rise to a transient thermal damping effect on microsecond time scales in sufficient thick (μm range) films[46]. However, this effect, which becomes undetectable in films as thin as 200 nm, can be reasonably neglected in practical thermometry applications. Finally, it should be stressed that these potential artifacts could be lowered by reducing the film thickness. In this case, the possible resulting loss of optical contrast could be counter-balanced by implementing a more sensitive (lock-in or inter-ferometry) detection system and by choosing the most suitable probe wavelength in the UV–Vis–NIR spectral range.

**Figures of merit of our SCO-based nanothermometer.** Using the above-discussed methodology, we have thoroughly assessed the achievable thermal, temporal and spatial resolution, which all constitute critical parameters for a fully fledged thermometer.

To assess the achievable thermal resolution, we performed a series of measurements to determine the temperature rise in a given 1-μm-wide area of the glass substrate (in the vicinity of the nanowire) for three close values of the applied electrical current ($I = 3.5, 3.6,$ and 3.7 mA). As depicted in Fig. 5a, b, a measurable temperature shift of about 1 °C is evidenced between the ON-state curves recorded for the three values of the applied current, and temperature rises of $22.3 \pm 0.2$, $23.3 \pm 0.2$, and $24.9 \pm 0.2$ °C are measured, respectively. These measurements demonstrate that the detection of temperature variations as small as 1 °C is possible on micrometer-size areas, using a simple CCD-based optical detection system. In any case, the overall thermal accuracy is limited to ca. 1 °C due to the existence of a narrow (0.5 °C-wide)

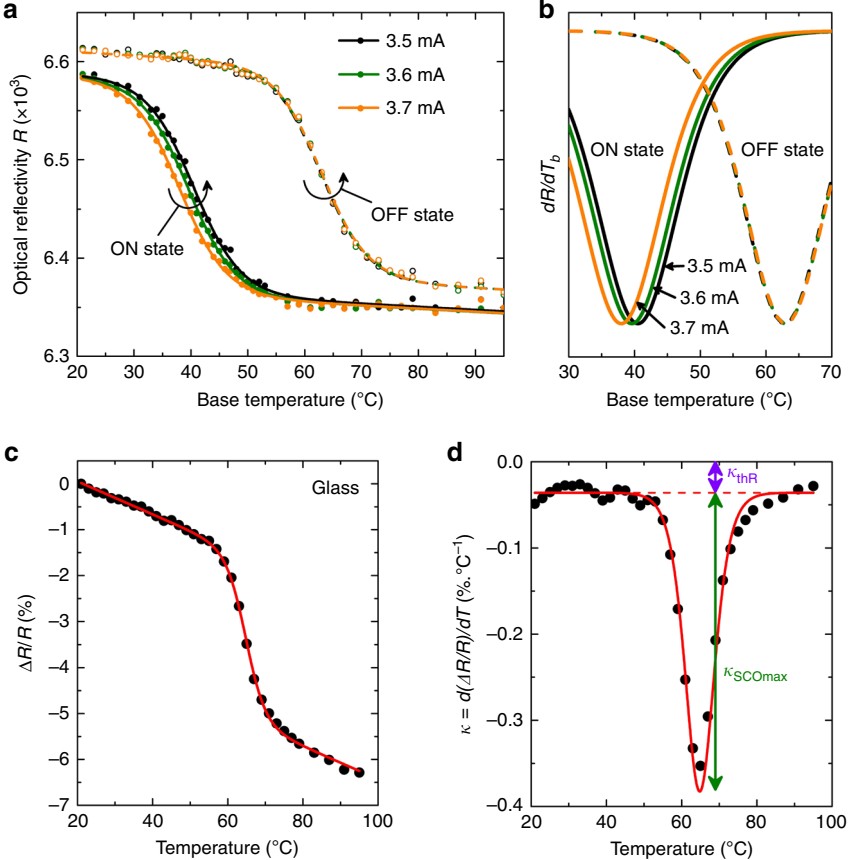

**Fig. 5 Temperature resolution and sensitivity. a** ON-state and OFF-state thermal transition curves recorded by optical reflectivity in a 1-μm-wide area of the SCO-coated glass substrate for three close values of the applied electrical current ($I = 3.5$, 3.6, and 3.7 mA). **b** Fitted derivative curves ($dR/dT_b$ vs. $T_b$) showing a measurable temperature shift for the ON-states. These measurements demonstrate that surface temperature variations as small as 1 °C can be unambiguously determined. **c** Typical thermal evolution of $\Delta R/R$ ($\lambda = 452$ nm) measured on a glass substrate coated with a 200-nm-thick film of **1**. **d** Thermal evolution of the corresponding thermoreflectance coefficient defined as $\kappa = d(\Delta R/R)/dT$. At the spin-transition temperature, the thermal sensitivity is found one order of magnitude larger compared to the ordinary thermoreflectance. Red solid lines correspond to the best fit to the data using a sigmoidal function.

thermal hysteresis loop in the thin films of **1**, and because of the slight thermal disturbance caused by the presence of the SCO layer on the heating element. Compared to conventional thermoreflectance techniques, the presence of the SCO thin film entails a substantial increase of the optical contrast ($\Delta R/R$), which implies an enhanced thermal sensitivity and detection limit around $T_{SCO}$. Figure 5c, d illustrate, respectively, the typical temperature dependence of $\Delta R/R$ measured on a glass substrate coated with a 200-nm-thick film of **1**, and the thermal evolution of the corresponding thermoreflectance coefficient $\kappa = d(\Delta R/R)/dT$. Far from the spin-transition temperature, the linear variation of $\Delta R/R$, which is accounted for the ordinary thermal expansion and associated refractive index change, coincides with a thermoreflectance coefficient of $\kappa_{thR} = -0.036\%$ °C$^{-1}$. As shown in Fig. 5d, a sizeable increase of $\kappa$ is observed due to the spin transition. The thermoreflectance coefficient reaches a maximum value of $\kappa_{thR+SCO} = -0.35\%$ °C$^{-1}$ at the transition temperature $T_{SCO}$, i.e. the temperature sensitivity is found (at maximum) 10 times greater compared to the conventional thermoreflectance. A similar gain in terms of thermal sensitivity was evidenced on various surfaces.

Regarding the temporal resolution, our thermometer is intrinsically limited by the thermally induced switching dynamics of the SCO molecules. From recent femtosecond optical spectroscopy measurements[38], we know that the thermally activated spin-state switching in the thin films of **1**, governed by the LS ↔

HS intramolecular energy barrier, occurs within a few tens of nanoseconds. This time scale thus defines the intrinsic response time of the film, in terms of optical reflectivity change, to any variation of temperature. An additional delay between the temperature rise in the device and the response of the thermometer can originate from the slow propagation of heat within the SCO layer, whose thermal diffusivity is rather low ($D_T = 2.6 \times 10^{-7}$ m$^2$ s$^{-1}$)[47]. Both numerical simulations[32] and ultra-fast pump-probe measurements[38] have demonstrated that the thermalization of SCO (150–300 nm thick) films, following a sharp current step or an ultra-short laser pulse, is completed in <1 μs. Such a temporal resolution is adequate for probing a large variety of dynamical phenomena, such as heat diffusion processes or for detecting brief heating events. As an example, Fig. 6a displays the time evolution of the optical reflectivity signal ($\Delta R/R$), measured by a gated CCD camera using a conventional pump-probe approach, on a SCO-coated gold microwire, following the injection of sharp current pulses with durations ranging from 10 to 100 μs. (Supplementary Fig. 10 shows the associated reflectivity images.) Close to the spin-transition temperature ($T_b = 55$ °C), a sizeable increase of the optical reflectivity is observed on the wire, while the same experiment carried out at $T_b = 25$ °C (sufficiently far from $T_{SCO}$) only shows a barely perceptible increase of the reflectivity—especially for short excitations. This example once again demonstrates the noticeable gain in terms of (thermal) sensitivity arising from the SCO

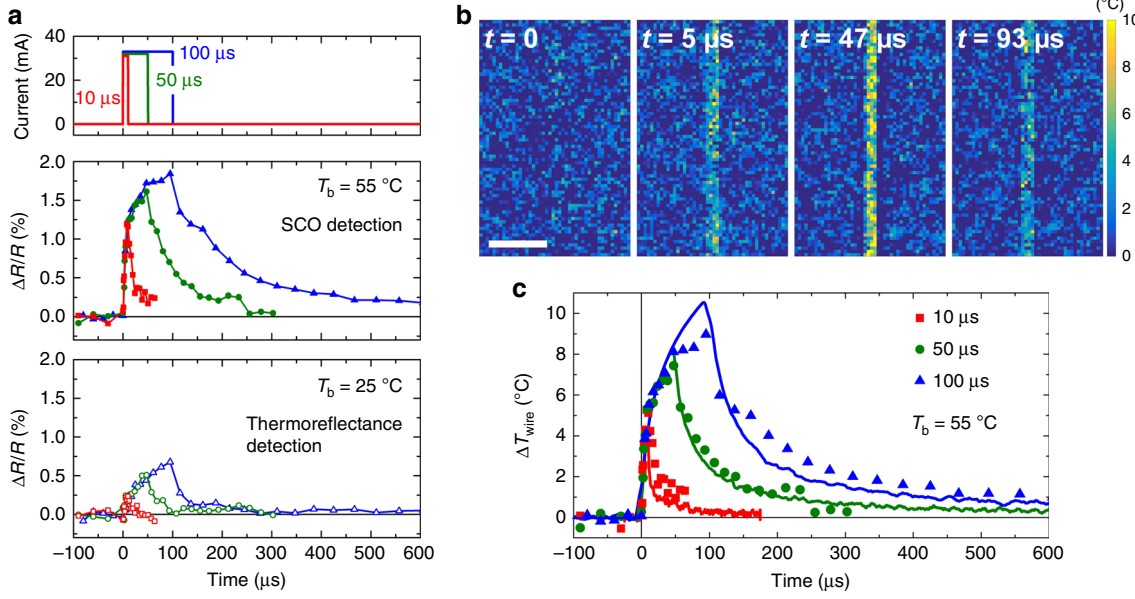

**Fig. 6 Temporal resolution and sensitivity. a** Time evolution of the optical reflectivity ($\Delta R/R$) measured on a SCO-coated gold microwire (1 mm × 8 μm × 300 nm) at two selected base temperatures, $T_b = 55$ °C and $T_b = 25$ °C, following the injection of sharp 10, 50, and 100-μs-long current pulses (32 mA). At 55 °C (close to the spin-transition temperature) the reflectivity change is amplified by the SCO phenomenon, while the measurements at 25 °C depict the ordinary thermoreflectance. **b** Temperature maps ($\Delta T$) deduced from the $\Delta R/R$ images at $T_b = 55$ °C at selected times, following a 50-μs-long current pulse. Scale bar, 40 μm. **c** Time evolution of the average temperature rise of the wire measured by our SCO-based thermometer. Solid lines correspond to the temperature profiles deduced from time-resolved electrical resistance measurements.

phenomenon. From the temperature evolution of the optical reflectivity measured in the OFF-state ($I = 0$), $\Delta R/R$ data acquired at 55 °C can be converted into $\Delta T$ values. Figure 6b depicts the temperature maps ($\Delta T$) obtained at selected times, following the application of a 50-μs-long current pulse. These measurements illustrate the remarkable achievement of imaging the surface temperature combining μm spatial and μs temporal resolution. In Fig. 6c, the time evolution of the average temperature rise measured on the wire by our SCO-based thermometer (data points) is compared to the temperature profiles deduced from time-resolved electrical resistance measurements[37] (solid lines) for the three durations of current pulses. As shown in Fig. 6c, the results from these two experiments are in good agreement.

In terms of spatial resolution, our SCO-thin-film thermometer, based on the measurement of the far-field optical reflectivity, involves a diffraction-limited, sub-μm spatial resolution. This was positively demonstrated through the measurements carried out on silicon, for which Joule heating could be well resolved on a 1-μm-wide wire (Fig. 4d). To further test the capabilities of our thermometer, we have also carried out a series of measurements on wires with truly nanometric dimensions on silicon substrates (Supplementary Fig. 11), in which the temperature distribution could be mapped down to 500-nm lateral resolution, but with a reduced signal-to-noise ratio. This spatial resolution obviously surpasses that achievable by IRT (see Supplementary Fig. 12), which remains the most conventional method of thermal imaging. To achieve even better spatial resolution, it is worthwhile to mention that the optical information could be also monitored using SPM techniques such as near-field scanning optical microscopy (SNOM)[32]. Besides, as the SCO phenomenon is also accompanied by a concomitant change of the mechanical, dielectric, and magnetic properties of the material, other approaches can be implemented to probe the thermal spin transition, instead of optical readout. Among them, quantitative AFM mechanical (Young's modulus) measurements[48,49] can be used to probe the temperature changes in the SCO-thin-film

thermometer, providing higher spatial resolution, though at the detriment of the temporal resolution and ease-of-use.

## Discussion

After decades of being laboratory curiosities with many potential applications, the development of this fully fledged surface nano-thermometer represents a notable breakthrough for the long-anticipated use of SCO molecules in actual technology. Similar to the history of photochromic molecules[50], the long-standing blocking point for SCO molecules has been their weak reliability and, in particular, their low endurance to cycling. Here, we have demonstrated reversible switching of a thin film of the molecular complex [Fe(HB(tz)$_3$)$_2$] over 10 million thermal cycles at elevated temperatures in ambient air. This discovery undeniably constitutes the decisive step forward real-world SCO-based applications and should promote fundamental studies to better understand the key structural parameters at the origin of the resilience/fatigability of SCO materials.

This excellent reversibility, together with the fast switching dynamics, strong optical contrast, reliable processing, and long-term stability, allowed us to turn this SCO film into a high-performance, nanoscale temperature sensor, which can be implemented on any surface. Using a simple CCD-based optical detection, we have operando validated the efficiency and accuracy of this SCO-based surface thermometer on various Joule-heated nanowires, in which heat distribution could be mapped down to sub-μm spatial, μs temporal, and 1 °C thermal resolution. Overall, this thermal-imaging technique appears as a versatile, cost-effective method, which has the advantage of simplicity for many demanding applications, ranging from sub-micron-level components to larger devices. Considering these appealing features, we believe that this approach could be readily incorporated as a simple, non-destructive test/diagnostic method into routine device-characterization and diagnostic protocols for a broad range of microelectronic, optoelectronic, and photonic devices.

# ARTICLE

## Methods

**Nanowire and microwire fabrication**. The gold nanowires were fabricated on top of glass and silicon wafers (Supplementary Fig. 5). The Si wafer was rendered insulating by thermal oxidation of its surface leading to the formation of an uppermost 600-nm-thick $SiO_2$ layer. The fabrication of the nanowires and the electrodes was divided into two steps. EBL was first used to deposit a 10 nm Ti/50 nm Au bilayer to form the metallic nanowires. The chips were then completed with an additional PL process enabling the deposition of a 10 nm Ti/300 nm Au bilayer to build the electrodes. The 10-nm-thick film of Ti serves as an adhesive layer between the Au and the $SiO_2$ passivation layer. Due to the insulating nature of the glass substrate, a 20-nm-thick layer of Ge was additionally deposited on top of the polymethylmethacrylate (PMMA) resist to form a conductive layer, indispensable to perform the e-beam nano-patterning. Due to their larger size, the microwires used for the thermal-cycling and time-resolved experiments were fabricated by means of conventional PL methods, enabling to pattern the wires and the electrodes at the same time[32].

**SCO sample synthesis and thin film deposition**. The bulk powder of **1** was synthesized as previously described[35]. The circuits were coated with a continuous 200-nm-thick film of **1**, grown by thermal evaporation at a base pressure of $2 \times 10^{-7}$ mbar. The bulk powder of **1** was heated until 250 °C in a quartz crucible and evaporated at a rate of 0.03 Å s$^{-1}$. The film thickness was monitored in situ using a quartz crystal microbalance and ex situ by atomic force microscopy (AFM) and through spectral reflectance measurements (Filmetrics, F20). The as-deposited films were recrystallized by a solvent–vapor annealing treatment, resulting in high-quality, homogeneous, oriented, nanocrystalline thin films[34].

**Experimental setup**. The SCO-coated chip was connected to an input current source-meter (Keithley 2611A), using an eight-track connector (see Supplementary Fig. 5), and mounted on a variable-temperature microscope stage (Linkam Scientific Instruments, LTS120) to control the base temperature ($T_b$) of the entire circuit. It is important to note that all the electrical characterizations on the nanowires and measurements were carried out by applying current and not voltage bias in order to guarantee the reproducible current excitation and resulting Joule heating. Optical reflectivity images were recorded using an Olympus BX51 upright microscope equipped with either a ×50 or ×100 magnification objective (numerical aperture, NA = 0.5 or 0.9, respectively) and a CCD camera (Andor Technology Clara, 1392 × 1040 pixels of 6.45 μm size). A ×0.5 lens was placed in front of the camera to increase the field of view. Time-resolved pump-probe reflectivity images were acquired using a gated Intensified-CCD camera (Andor Technology iStar DH734, 1024 × 1024 pixels of 13 μm size) triggered by a function generator (Tektronix, AFG3022C) allowing the generation of time-variable current pulses. The sample was illuminated by a halogen lamp, but the spectral range was reduced using a band-pass filter ($\lambda = 452 \pm 22$ nm). Infrared thermography (IRT) images of the Joule-heated nanowires were acquired within the 7.5–13 μm spectral range using a 640 × 480 pixels camera (Micro Epsilon thermoIMAGER TIM 640, instantaneous field of view of 28 μm). The transient temperature rise of the microwires was monitored, with a sub-μs time resolution, using a custom-made differential resistance measurement setup[37] (Supplementary Fig. 4), the output signal being captured with an oscilloscope (Tektronix, DPO3014).

**SCO sample characterization**. Temperature-dependent absorbance spectra of the films, deposited on fused silica substrates, were collected at wavelengths between 200 and 800 nm, using a Cary 50 spectrophotometer (Agilent Technologies) and a heating/cooling stage (Linkam Scientific Instruments, FTIR600). Spectra were acquired in the 20–120 °C range with a scan rate of 1 °C min$^{-1}$. AFM topography measurements were performed using a Cypher-ES microscope (Oxford Instruments) in amplitude-modulation mode in ambient air, using OMCL-AC160TS-R3 (Olympus) probes. Scanning electron microscopy (SEM) images of the metallic wires were recorded using a JSM-7800F Prime microscope (JEOL) operated at 5 kV.

**Finite-element simulations**. 2D and 3D temperature distributions in the heating wires were simulated using the finite-element method (FEM) by numerically solving the heat equation as implemented in the software package COMSOL. Due to the small heating area and the relatively narrow temperature range covered, the radiation losses were considered negligible. This enables a simplified thermal description of the system, which largely depends on the power input for the heat generation via Joule heating in the metallic wire, the thermal properties of the materials (see Supplementary Table 1) and the system boundary conditions for the power losses. The value of the electrical resistivity of gold ($1 \times 10^{-7}$ Ω m), which is the only adjustable parameter, was estimated through the mean electrical resistance of the nanowires measured at room temperature ($R_e \sim 160$ Ω). This value, which is larger than the bulk value ($2.4 \times 10^{-8}$ Ω m), was already reported in the literature[51]. Owing to the relatively small temperature changes in our experiments, the temperature dependence of the material properties was not taken into account in the simulations.

## Data availability

The authors declare that all data supporting the findings of this study are available within the paper and its supplementary information files.

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

## Acknowledgements

We are grateful to Franck Carcenac and Mario Piedrahita-Bello for help with EBL and SCO sample synthesis. This work was supported by the European Commission through the SPINSWITCH project (H2020-MSCA-RISE-2016, Grant Agreement No. 734322), and by the Micro and Nanotechnologies Platform of LAAS-CNRS (Toulouse, France), which is a member of the French RENATECH Network. The Ph.D. of Y.Z. is supported by the China Scholarship Council.

## Author contributions

A.-C.B. and Y.Z. realized the spin-crossover thin film deposition and characterization. K.R. performed the thermometry experiments and analyzed the data under the supervision of G.M. and A.B., and discussion with L.R., L.S., and C.B. The finite-element simulations were performed by C.B. The manuscript was written by K.R. and G.M. with comments and input from all the authors.

## Competing interests

A patent covering parts of the manuscript has been filed by the authors.
