## [Peer Review File · Nature Communications]

Reviewers' Comments:

Reviewer #1:

Remarks to the Author:

The manuscript titled "Unprecedented switching endurance affords for high-resolution surface temperature mapping using a spin-crossover film" demonstrated the surface temperature measurement with diffraction-limited sub- μm spatial, μs temporal and 0.5 $^{\circ}\text{C}$ thermal resolution. This is a solid piece of work and is overall qualified in Nature Communications, while some problems need to be clarified before publication.

1. The molecular complex $[\text{Fe}(\text{HB}(1,2,4\text{-triazol-1-yl})_3)_2]$ has a spin-transition property and high switching endurance. The performance is outstanding. Maybe the authors can add some comments on the design principle of the SCO compound used in this work, or add some comparison between this structure and previously reported ones.
2. The Joule-heated gold nanowire is relatively smaller than the thermal sensing film. When a small amount of heat is produced in the nanowire in a non thermal equilibrium situation, this heat is conducted through the film, causing a temperature artifact. The authors should consider the heat production of the nanowire, the conduction property and heat capacity of the nanowire and the thermal sensing film.
3. One of the advantages of optical thermometry is that it is non-invasive and remote. The thin film demonstrated here still remains to be a contact thermometry, and the interference on the film surface would clearly affect the temperature measurement. The authors should give some outlook on future development of the thermal sensing film addressing the problems mentioned above.

Reviewer #2:

Remarks to the Author:

I find that figure 1 is perhaps the most impressive part of this paper. Usually, with organic, or even metal organic devices, there might 1, 2 or 3 trials. If the investigators are very diligent, maybe 100 trials. Here the authors have done 10 millions trials. Importantly, they have reproducibility without degradation. This is an extremely important insight as this speaks to applicability. Nice to have device that works, and is reproducible, but these SCO films are robust to an extent I have never before seen reported. The spatial resolution of the temperature map, as demonstrated in Figures 3 and 4, is also very impressive.

If I have a criticism it that because of the need to calibrate the reflectivity, which appears to be subject to variations in SCO film thickness and other experimental variations, combined with the fact that the change in reflectivity is only about 10% of the total reflectivity, I less certain about the claims of the thermal accuracy of this type of thermometry. I do not think this is a selling point of this, otherwise, outstanding experimental effort, and would recommend a more cautious claim than thermal accuracy to ± 0.5 degrees unless a more compelling case can be made - Figure 4a tends to detract from this claim somewhat.

Reviewer #3:

Remarks to the Author:

Title: Unprecedented switching endurance affords for high-resolution surface temperature mapping using a spin-crossover film

Non-invasive precise thermometers working at the nanoscale with high spatial resolution, where the conventional methods are vain, have appeared over the last decades as a very active field of exploration. This has been highly inspired by the numerous challenging requests arising from nanotechnology and biomedicine. So far, various materials such as luminescent (organic dyes, QDs, and Ln^{3+} ions as thermal probes, as well as more complex thermometric systems formed by polymer and organic-inorganic hybrid matrices encapsulating these emitting centers) and non-luminescent thermometers (including scanning thermal microscopy and other thermometers (e.g., nanolithography, CNT thermometry, and biomaterials thermometry) have been introduced at nanometric scale [Nanoscale, 2012,4, 4799-4829].

For the present report, I appreciate the achievements made on high-resolution surface temperature mapping using a spin-crossover (SCO) film with high durability (over 10 million thermal cycles) and endurance. I think the work is considered as novel owing to its high improvement in the reliability of the SCO molecule. On the other hand, the previously developed

photochromic molecules are also displayed remarkable achievements like strong optical contrast with fast/moderate switching dynamics in which the authors presented as high achievements. Further, several alternative strategies have also been designed so far on SCO as the authors documented since then, including a review on molecule-based spin-crossover materials [Chem. Soc. Rev., 2011,40, 3313-3335] and other investigations [Chem. Commun., 2012,48, 4163-4165; Dalton Trans., 2019,48, 16853-16856; J. Mater. Chem. C, 2020, DOI: 10.1039/D0TC01532F]. Meanwhile, the integration and identifiable solid-solid phase transition performance of the phase change material is not clear most likely due to the absence of differential calorimetric measurements. Appreciating the initiation and efforts made for the work, I think the present work is not suitable to consider in Nature communication. I hope the paper will be considered to other reputable journals.

Reviewer #1 (Remarks to the Author):

The manuscript titled "Unprecedented switching endurance affords for high-resolution surface temperature mapping using a spin-crossover film" demonstrated the surface temperature measurement with diffraction-limited sub- μm spatial, μs temporal and $0.5\text{ }^\circ\text{C}$ thermal resolution. This is a solid piece of work and is overall qualified in Nature Communications, while some problems need to be clarified before publication.

1. The molecular complex $[\text{Fe}(\text{HB}(1,2,4\text{-triazol-1-yl})_3)_2]$ has a spin-transition property and high switching endurance. The performance is outstanding. Maybe the authors can add some comments on the design principle of the SCO compound used in this work, or add some comparison between this structure and previously reported ones.

Reply: The relation between the molecular structure and the outstanding resilience of compound **1** is still an open question. In any case, this exceptionally high switching endurance must arise from a **combination of structural factors**. Among them, the **isostructural character** of the spin transition in **1** (same orthorhombic space group *Pbca* in the two phases), the **lack of strong intermolecular bonds** (as opposed to SCO coordination networks for example) as well as the **pronounced anisotropy of the transformation strain** are all important features allowing to minimize the transformational mechanical stress and thus to better accommodate the volume change associated with the spin transition. In particular, the volume expansion in **1** mainly takes place along the *c*-axis ($\Delta c/c = +5.6\%$) [*CrystEngComm*, **2017**, 19, 3271]. Since thin films of **1** are oriented (the *c*-axis being normal to the substrate), **the transformation strain is almost zero in the substrate plane** ($+0.17\%$) [*J. Am. Chem. Soc.*, **2018**, 140, 8970]. Hence, despite the relatively large volume strain (ca. 4.5%), the spin transition gives rise to rather small mechanical stress in the film. More generally, an anisotropic structural deformation can lead to the emergence of **mismatch-free directions during the phase transformation**, allowing to channel a part of the internal stresses/frictions. For example, in the SCO compound $[\{\text{Fe}(\text{NCSe}(\text{py})_2)_2(\text{m-bpypz})\}]$, the observation of a preferential orientation of the LS/HS phase boundaries during the spin-state transformation was correlated to the existence of such mismatch-free directions [*Angew. Chem. Int. Ed.*, **2014**, 53, 7539]. On the contrary, molecular crystals in which the molecules are interlocked with (quasi)isotropic interactions are known to be more brittle [*CrystEngComm*, **2010**, 12, 2296]. Overall, we believe that the exceptionally high switching endurance of thin films of **1** arises from the combination of these factors.

To address this point, we have added/amended the following on pages 4 and 5 of the manuscript:

“The exceptionally high switching endurance of thin films of **1 must probably arise from a combination of structural factors. First, the molecular nature of compound **1** (with only weak intermolecular bonds) and its processing in the form of high-purity (vacuum-deposited) thin films are undoubtedly important features that minimize the transformational stress during the spin-state switching. Most importantly, as demonstrated for shape memory alloys^{40,42}, and also suggested for VO_2 ⁴³, the overall crystallographic compatibility of the two phases appears as a key parameter to achieve high switching endurance. In **1**, this compatibility includes the isostructural character of the transition (same orthorhombic space group *Pbca* in the two phases) as well as a pronounced anisotropy of the transformation strain, the structural deformation mainly occurring along the *c*-axis ($\Delta c/c = +5.6\%$)³⁵. This anisotropic deformation allows the crystal lattice to better accommodate the volume change, through the possible formation of mismatch-free LS/HS phase boundaries during the spin-state switching⁴⁴. On the other hand, as the thin films of **1** are oriented (with the *c*-axis normal to the substrate), the**

transformation strain is almost zero in the substrate plane⁴⁵ and the spin transition thus gives rise to rather small mechanical stress in the film.”

2. The Joule-heated gold nanowire is relatively smaller than the thermal sensing film. When a small amount of heat is produced in the nanowire in a non-thermal equilibrium situation, this heat is conducted through the film, causing a temperature artifact. The authors should consider the heat production of the nanowire, the conduction property and heat capacity of the nanowire and the thermal sensing film.

Reply: The reviewer is right. The presence of the SCO thin film will introduce some disturbance to the temperature field. Nevertheless, under quasi-static conditions, this effect remains marginal in most cases. Indeed, our finite-element analysis has demonstrated that the surface temperature differs by less than 1 °C due to the presence of the SCO layer (see **Figure S9** in Supporting Information), which remains thus comparable with the overall measurement uncertainty. The parameters used for the simulations (thermal conductivity and heat capacity of the nanowire, the SCO film and the substrate) are listed in **Table 1** (page 12). In particular, considering the thermal diffusivity of the SCO material ($D = 2.6 \times 10^{-7} \text{ m}^2 \cdot \text{s}^{-1}$ [*Phys. Rev. B*, **2017**, 96, 134106]), the characteristic time associated with heat diffusion across the film of thickness $e = 200 \text{ nm}$ is $\tau = e^2 / D = 0.15 \text{ } \mu\text{s}$. It means that the static regime is typically reached within this time scale. This is corroborated by time-resolved optical measurements [*Adv. Mater.*, **2019**, 31, 1901361], which show that the thermalization of our SCO films is completed in less than 1 μs . This time scale thus defines the achievable temporal resolution.

We have completed the part of the manuscript which deals with this issue (page 7):

“We took benefit from these finite-element simulations to assess the possible disturbance in the temperature field caused by the presence of the SCO layer. Under (quasi)static conditions – typically reached within the microsecond time scale (*vide infra*) – we find that the surface temperature differs by less than 1 °C due to the presence of the SCO layer (Figure S9). This value remains comparable with the overall measurement uncertainty.”

3. One of the advantages of optical thermometry is that it is non-invasive and remote. The thin film demonstrated here still remains to be a contact thermometry, and the interference on the film surface would clearly affect the temperature measurement. The authors should give some outlook on future development of the thermal sensing film addressing the problems mentioned above.

Reply: We agree with the reviewer. For the reasons mentioned in our previous answer, our SCO-based thermometer – as all thermometry methods based on surface coating – can be reasonably classified as a semi-invasive technique. The simplest way to limit the impact of the SCO thin film on the temperature measurement is to reduce the film thickness. However, the reduction of the thickness could result in a loss of the optical contrast (*i.e.* of the thermal sensitivity). This effect could be counterbalanced by implementing a more sensitive detection system (optimized lock-in or interferometry system for example), or by exploring different spectral ranges, because the change of the refractive index implies an optical response over the entire UV-Vis-NIR spectral range.

We have added a note (page 8) to address this specific point:

“Finally, it should be stressed that these potential artefacts could be lowered by reducing the film thickness. In this case, the possible resulting loss of optical contrast could be counterbalanced by implementing a more sensitive (lock-in or interferometry) detection system and by choosing the most suitable probe wavelength in the UV-Vis-NIR spectral range.”

Reviewer #2 (Remarks to the Author):

I find that figure 1 is perhaps the most impressive part of this paper. Usually, with organic, or even metal organic devices, there might be 1, 2 or 3 trials. If the investigators are very diligent, maybe 100 trials. Here the authors have done 10 million trials. Importantly, they have reproducibility without degradation. This is an extremely important insight as this speaks to applicability. Nice to have a device that works, and is reproducible, but these SCO films are robust to an extent I have never before seen reported. The spatial resolution of the temperature map, as demonstrated in Figures 3 and 4, is also very impressive.

If I have a criticism it is that because of the need to calibrate the reflectivity, which appears to be subject to variations in SCO film thickness and other experimental variations, combined with the fact that the change in reflectivity is only about 10% of the total reflectivity, I am less certain about the claims of the thermal accuracy of this type of thermometry. I do not think this is a selling point of this, otherwise, outstanding experimental effort, and would recommend a more cautious claim than thermal accuracy to ± 0.5 degrees unless a more compelling case can be made - Figure 4a tends to detract from this claim somewhat.

Reply: The level of the reflectivity signal can effectively change depending on various experimental conditions, but, precisely, the main strength of our method is that the determination of the temperature is based on the reflectivity jump (threshold detection) occurring at the transition (taking advantage of the abrupt character of the spin transition), and NOT on the percentage change in reflectivity. As a consequence, surface temperature can be determined, in principle, without any thermal calibration of the reflectivity signal. On the other hand, as correctly noted by the reviewer, a thermal resolution of ~ 0.5 °C has been demonstrated (Figure 4a) in rather specific conditions and at the cost of considerable experimental effort. This value is certainly the ultimate achievable temperature resolution. Indeed, in any case, the accuracy is limited by the existence of a 0.5-°C-wide thermal hysteresis loop in the thin films of **1**, and by the thermal disturbance caused by the presence of the SCO layer on the heating element (see question 2 of reviewer #1). To be more general, following the recommendation of the reviewer, it is therefore more reasonable to consider a thermal accuracy of the order of ± 1 °C. **This claim has been corrected in the whole manuscript.**

We have also added the following note on page 8:

“In any case, the overall thermal accuracy is limited to *ca.* 1 °C due to the existence of a narrow (0.5-°C-wide) thermal hysteresis loop in the thin films of **1, and because of the slight thermal disturbance caused by the presence of the SCO layer on the heating element.”**

Reviewer #3 (Remarks to the Author):

Non-invasive precise thermometers working at the nanoscale with high spatial resolution, where the conventional methods are vain, have appeared over the last decades as a very active field of exploration. This has been highly inspired by the numerous challenging requests arising from nanotechnology and biomedicine. So far, various materials such as luminescent (organic dyes, QDs, and Ln³⁺ ions as thermal probes, as well as more complex thermometric systems formed by polymer and organic-inorganic hybrid matrices encapsulating these emitting centers) and non-luminescent thermometers (including scanning thermal microscopy and other thermometers (e.g., nanolithography, CNT thermometry, and biomaterials thermometry) have been introduced at nanometric scale [Nanoscale, 2012,4, 4799-4829].

For the present report, I appreciate the achievements made on high-resolution surface temperature mapping using a spin-crossover (SCO) film with high durability (over 10 million thermal cycles) and endurance. I think the work is considered as novel owing to its high improvement in the reliability of the SCO molecule. On the other hand, the previously developed photochromic molecules are also displayed remarkable achievements like strong optical contrast with fast/moderate switching dynamics in which the authors presented as high achievements. Further, several alternative strategies have also been designed so far on SCO as the authors documented since then, including a review on molecule-based spin-crossover materials [Chem. Soc. Rev., 2011,40, 3313-3335] and other investigations [Chem. Commun., 2012,48, 4163-4165; Dalton Trans., 2019,48, 16853-16856; J. Mater. Chem. C, 2020, DOI: 10.1039/D0TC01532F].

Reply: Indeed, many papers have been published on the molecular spin-crossover phenomenon and its potential applications ... The state-of-the-art has been recently reviewed by us [*Adv. Mater.* **2018**, 30, 1703862] and others [*Coord. Chem. Rev.* **2017**, 346, 176]. However, in all of the previously reported studies, the SCO cycling was only restricted to few/tens/hundreds thermal cycles before failure. This issue has constituted so far the long-standing blocking point for SCO molecules, and researchers – including ourselves (!) – had many doubts about the real applicability of these switchable molecules.

The problem of fatigue is not only central for the spin-crossover field, but also for the vast field of molecular switches. We can cite here the first phrases of a recent paper of Ben Feringa [*J. Phys. Chem. C* **2019**, 123, 25908]: “The stability and longevity of the device are often limited by the robustness of the [molecular] switches which, in practice, tend to fatigue after only a few switching cycles, particularly when immobilized on a surface. The most common photoswitches - azobenzene, dithienylethenes, and spiropyrans - all suffer various types of fatigue: photochemical fatigue, decomposition in a reactive environment (e.g., oxidation), and inter/intramolecular side-reactions.”

Here, we come up with an industrial quality molecular film, which we are able to (re)synthesize with better than 1 °C accuracy of the transition temperature from one batch to another, which withstands more than 10 million endurance cycles in ambient air, and which allows, for the first time, the development of a truly competitive, fully-fledged SCO-based technological application. The unprecedented features reported here, have definitely removed our doubts about the applicability of these compounds ... and we feel through the reaction of the first two reviewers that our enthusiasm will be shared by the readers. We are convinced that the cornerstone we reach in this work, will motivate substantial new research in the field of molecular materials!

We have slightly amended the introduction (pages 2 and 3) to reinforce this message:

“Although first proofs of concept have been reported^{32,33}, the realization of a real-world SCO-based application was primarily hampered by the weak reliability and the recurrent problem of mechanical fatigue of SCO materials upon extended switching, the SCO cycling being typically restricted to only few/tens/hundreds thermal cycles before failure. This issue has constituted so far, the long-standing blocking point for SCO molecules.

In this study, we show that the molecular SCO complex [Fe(HB(tz)₃)₂] (tz = 1,2,4-triazol-1-yl) **1, recently synthesized in the form of high-quality, vacuum-deposited thin films³⁴, overcomes this major bottleneck.”**

Meanwhile, the integration and identifiable solid-solid phase transition performance of the phase change material is not clear most likely due to the absence of differential calorimetric measurements. Appreciating the initiation and efforts made for the work, I think the present work is not suitable to consider in Nature communication. I hope the paper will be considered to other reputable journals.

Reply: The solid-solid phase transition performance of our films has been established using calorimetric measurements: the enthalpy change associated with the spin transition is $\Delta H = 40 \text{ J.g}^{-1}$ [*CrystEngComm*, **2017**, 19, 3271]. This value is much lower than the melting enthalpy of solid-liquid phase-change materials ($\Delta H > 150 \text{ J.g}^{-1}$), and in the average among the solid-solid phase-change materials [*Appl. Therm. Eng.* **2017**, 127, 1427]. Of course, this excess heat capacity can give rise to a transient thermal damping effect on microsecond time scales in thick (μm range) films [*Adv. Mater.*, **2020**, 2000987]. However, this effect becomes negligible in films as thin as 200 nm.

We have added the following note to the revised manuscript (pages 7-8) on this issue:

“Obviously, under non-thermal-equilibrium conditions, the temperature difference between the heating element and the thin-film surface is expected to be larger. In particular, in addition to the “normal” heat capacity of the SCO film, an excess heat capacity should be also considered due to the endothermic character of the LS-to-HS transition ($\Delta H = 40 \text{ J.g}^{-1}$ in 1)³⁵, which can give rise to a transient thermal damping effect on microsecond time scales in sufficient thick (μm range) films⁴⁶. However, this effect becomes undetectable in films as thin as 200 nm and can be reasonably neglected in practical thermometry applications.”

Reviewers' Comments:

Reviewer #1:

Remarks to the Author:

The authors have revised the manuscript according to the reviewers' comments. Now it is acceptable.

Reviewer #2:

Remarks to the Author:

I think the paper is more than acceptable for publication.

Reviewer #3:

Remarks to the Author:

It can be published. All my comments and concerns are addressed in the revised Article.

We thank the time and consideration of the reviewers.

REVIEWERS' COMMENTS:

Reviewer #1 (Remarks to the Author): The authors have revised the manuscript according to the reviewers' comments. Now it is acceptable.

Reviewer #2 (Remarks to the Author): I think the paper is more than acceptable for publication.

Reviewer #3 (Remarks to the Author): It can be published. All my comments and concerns are addressed in the revised Article.